# Nutritional Viewpoints on Eggs and Cholesterol

**DOI:** 10.3390/foods10030494

**Published:** 2021-02-25

**Authors:** Michihiro Sugano, Ryosuke Matsuoka

**Affiliations:** 1Kyushu University, Fukuoka 819-0395, Japan; suganomi@deluxe.ocn.ne.jp; 2Prefectural University of Kumamoto, Kumamoto 862-8502, Japan; 3Japan Egg Science Society, Tokyo 182-0002, Japan; 4R&D Division, Kewpie Corporation, Tokyo 182-0002, Japan

**Keywords:** egg, cholesterol, cardiovascular disease, coronary heart disease, stroke, diabetes mellitus, lifestyle, food matric, dietary pattern, Japanese characteristics

## Abstract

Although most current epidemiologic studies indicate no significant association between consuming one egg daily and blood cholesterol levels and cardiovascular risk, arguments still persist with a positive association. Since the diet is one of the most influential factors for this association, we illustrate characteristic features in Japanese people whose dietary pattern is distinct from that, for example, the US (United States) population. Available epidemiologic studies in healthy Japanese people show no association between consumption of one egg daily and blood cholesterol level, consistent with those observed in the US population. However, when consumption of major nutrients and food sources of cholesterol are compared to the US population, Japanese people may have an extra-reserve against the influence of eggs on cardiovascular risk markers, despite consuming relatively more eggs. Further discussion on the influence of nutrients contained in the egg and dietary pattern, including interaction with gut microbes, is necessary. In addition, special consideration at the personalized level is needed for judgment regarding dietary cholesterol not only for hypercholesterolemic patients but for hyper-responsive healthy persons. Although randomized controlled trials with long-term follow-up are required to evaluate the association between consumption of eggs and human health, available information, at least from the nutritional viewpoint, suggests that egg is a healthy and cost-efficient food worldwide.

## 1. Introduction

The widely-held belief to restrict cholesterol intake was recently removed from several dietary guidelines worldwide, and only a proviso describing care for hypercholesterolemic patients remains [1,2]. As a result, moderate egg intake can be a part of a healthy diet. Since this transition depends on the results of epidemiologic studies, the judgment cannot be generalized and is limited only to the participants of the corresponding cohorts. In addition, the response pattern to egg consumption possibly differs between people living in different countries, as characterized in the risk of type 2 diabetes mellitus (T2DM) in the US (United States) population [3,4].

Although careful judgment is unavoidable for understanding the association between dietary cholesterol and blood cholesterol levels, there is confusion in terms of understanding the new guidelines. To evaluate the effects of eggs on blood cholesterol levels, it is essential to consider the kind of foods simultaneously consumed. Eggs are rich in cholesterol but also contain components capable of interfering with cholesterol absorption from the intestine and disconnecting with a rise of blood cholesterol [5,6,7]. This is in contrast to the case of beef meats that contain both cholesterol and saturated fatty acid (SFA) at relatively high levels.

The US FDA (Food and Drug Administration) presently does not classify eggs as healthy because they are too high in fat and cholesterol, despite a relatively lower SFA content (FDA food labeling regulations, “healthy” label claims). The restriction of cholesterol intake still exists in the dietary guidelines of many countries [8]. In clinical settings, restriction of cholesterol intake has been long-standing the principal recommendation [9]. The safety theory on cholesterol is, therefore, not broad-certified, and judgment varies among the commentators concerned. An expert engaged in the field of egg production has reported that eggs are no longer unsafe [10], while a doctor insists that a report that cholesterol intake does not have a serious influence on low density lipoprotein (LDL)-cholesterol, the risk of cardiovascular disease (CVD) should be considered conditional, and it is appropriate to refrain from egg consumption [11]. A researcher in the nutrition field has stated that to be fair, the answer to the question as to whether eggs are bad is probably, “no” [12]. Under these conditions, it is unwise to entrust judgment to consumers, and the lingering concept of a rumor about cholesterol cannot be helpful.

Notably, as eggs contribute approximately half of the total dietary cholesterol intake in Japan, it is reasonable that researchers in clinical settings remain to consider eggs as a risk. The same concept is true in the US, although eggs are not major contributors to dietary cholesterol (Figure 1) [13]. A large-scale, long-term randomized controlled trial (RCT) is essential to draw the definite answer to this issue; however, it is practically difficult to perform such a study mainly due to ethical standpoints. The results of observational studies are limited to draw definite conclusions. Therefore, the debate on the judgment of whether eggs are friends or foes continues even in Japan, since the lifetime risk for coronary heart disease (CHD) due to the presence of hypercholesterolemia is evident, especially in men [14].

This review focuses on the effect of egg consumption on blood cholesterol levels, and hence, CVD risk in Japanese whose regular diet comprises of Washoku (UNESCO recognized Japanese cuisine as intangible cultured heritage), an example of a healthy diet, similar to the Mediterranean diet, to know how a lifestyle in a particular dietary pattern modifies this issue. This line of investigation will contribute to understanding the association between diets and CVD risk and will provide a reference from universal to a personal level. However, many Japanese recognize that egg consumption results in elevation of blood cholesterol level and causes CHD. Moreover, a recent epidemiologic study indicates that blood cholesterol levels are increasing in East Asian countries, including Japan, in contrast to a lowering trend in Western countries [15].

When considering the practical difficulty of performing high-quality RCTs with the egg, it is imperative to entrust available well-controlled prospective observational or intervention studies that emphasize the relations with dietary patterns. However, unlike in the US, food and drugs are managed by different ministries in Japan, such as foods in the Ministry of Agriculture, Forestry and Fisheries, and the medicinal drugs in the Ministry of Health, Labor, and Welfare. Since the latter ministry is responsible for establishing dietary guidelines, the advisory members in the dietary reference intakes for Japanese are constituted mostly by medical doctors, and the specialists in the field of nutrition and food science are limited. As a result, it is quite astonishing that clinicians, who typically did not learn about nutrition as medical students [16,17,18], play a key role in the dietary guidelines, and therefore, a judgment is apt to be inclined to a clinical viewpoint in Japan. Under these situations, there is no limit on the debate whether eggs are a friend or a foe.

## 2. Egg Consumption and Response of Blood Cholesterol in Japanese

The relationship between egg consumption and blood cholesterol level in Japanese can be read in the NIPPON DATA, a series of observation studies in the National Integrated Project for Prospective Observation of Noncommunicable Diseases and the Trends in the Aged, supported by the Ministry of Health, Labor, and Welfare of Japan. In NIPPON DATA80 (1980) [19], there was a significant positive association between egg consumption and blood cholesterol level in women, but not in men. However, the association disappeared in the NIPPON DATA90 (1990) [20], although the blood cholesterol levels increased above those observed in prior studies in both sexes (Figure 2) [19,20,21]. The increase in blood cholesterol level in the latter study is considered to reflect the increase in cholesterol consumption from foods other than the egg, according to changes in eating patterns during this decade. The trend was duplicated in NIPPON DATA 2010 [22], and there was no association between egg consumption and blood cholesterol [22]. These observations may provide the importance of a dietary pattern in the regulation of blood cholesterol levels. On the other hand, the results of a Japan public health center (JPHC)-based prospective study (1990–2001) indicated the association between serum cholesterol concentration and CHD, but not between the number of eggs consumed and CHD [23]. From these observation studies, it is difficult to draw a conclusion between egg consumption and blood cholesterol level. The NIPPON DATA 90 also showed no association between the daily intake of one egg and CVD death both in men and women, but in the intake of more than two eggs per day that showed a significant increase in the total mortality and cancer death in women, but not in men [19]. However, the authors of these epidemiologic studies consistently insist on limiting egg consumption to some extent for health. In a population-based, random-sample, cross-sectional study among Japanese people aged 40–59 years (INTERLIPID, 1969–1999), a higher intake of dietary cholesterol correlated with a higher concentration of the serum LDL-cholesterol only in males, who are less educated and unemployed, whereas the contrary was observed in educated and employed males [23]. These results show the complexity of the relationship between egg consumption and blood cholesterol level in Japan. The same group of researchers re-evaluated the association of egg intake with serum total-cholesterol level and cause-specific and total mortalities in Japanese women and found that egg intake was not associated with CVD mortality but was associated with cancer and total mortality [24].

This study concluded that reducing egg intake may have some definite health benefits for women in Japan.

Information from randomized intervention trials with Japanese people is limited, but available data do not show a positive association between egg consumption and blood cholesterol levels. In one study in which 750 mg of cholesterol as egg yolk was given to healthy and hyperlipoproteinemic subjects for four weeks, there were no changes in serum LDL- and high density lipoprotein (HDL)-cholesterol levels [25]. However, serum cholesterol was not elevated in total but elevated in one-third of participants, who are hyper-responders (the proportion of hyper-responders is comparable with that reported in studies abroad [26]). Thus, attention to hyper-responders is disregarded. In a series of small-scale intervention trials in healthy and high blood cholesterol adults, an addition of one egg daily to ordinary diets for four weeks did not influence serum total-cholesterol levels [27,28,29], while there were trends of increasing HDL-cholesterol and decreasing LDL-cholesterol/HDL-cholesterol ratio in one study [27]. In addition, dietary eggs prevented the oxidation of LDL and improved participants’ nutritional state in all studies. The proportion of hyper-responders was not described.

Although the relationship between egg consumption and blood cholesterol level in Japan may only be understood from available information, it seems appropriate to judge that daily consumption of one egg does not influence the blood cholesterol level and hence, probably a CVD risk in healthy Japanese adults eating balanced diets. There were at least no unhealthy changes in blood cholesterol levels and cholesterol distributions in different lipoproteins in available intervention studies in Japan. The recent observation shows an increasing trend of serum total-cholesterol levels in Japan since 1980, as shown in Figure 3, but this was reflected by an increase in HDL-cholesterol and the decreased ratio of the total- and HDL-cholesterol [30]. This trend was more remarkable in women than in men. A similar but moderate changing trend was also duplicated in South Korea [30], and a recent study showed the association between serum LDL-cholesterol and the amount of dietary cholesterol [31]. There has been an increasing trend in total-cholesterol levels in China more than in Japan, which has been attributed to the increase in both non-HDL-cholesterol and the ratio of the total- and HDL-cholesterol [30]. The changing pattern of blood cholesterol differs among these three East Asian countries, suggesting that the specific response pattern of Japanese probably reflects a difference in the dietary pattern among these countries. Compared to a decreasing trend of both total- and non-HDL-cholesterol in Western countries, the changing pattern of blood cholesterol in Japan is quite specific. The current decreasing trend of mortality due to heart disease in Japan may at least in part be related to the changes in the lipoprotein cholesterol level, although its lifetime risk in Japan is high in people high in blood cholesterol [32].

## 3. Egg and Health: Comparison between Japan and the US

There was a detectable difference in the major death cause between Japan and the US population, though CVD and cancer are the top two mortalities in both countries (death rate for cancer and CVD is 31.9% and 21.5% for Japan, and 21.9% and 31.7% for the US, respectively, United Nations-World Population Prospects). Blood cholesterol level is somewhat higher in Japanese than in the US population, but CHD mortality is lower in Japanese. Therefore, systematic estimation of the Japanese diet is of interest in relation to egg consumption, blood cholesterol level, and CVD risk.

To understand the response of blood cholesterol level to egg consumption, special consideration to the dietary pattern and the number of eggs consumed is needed. As shown in Table 1, the current nutrient intake in Japanese and US adults differs considerably with respect to major nutrients. A higher level of total energy intake in US adults is probably reflecting the difference in body size compared to Japanese, but the difference in the amounts of dietary fats is beyond this criterion. The composition of dietary fatty acids also differs, particularly with more SFA in the US. Although US adults are consuming almost twice as many polyunsaturated fatty acids (PUFA), specifically n-6 PUFA linoleic acid, compared to Japanese, the proportion of n-6 to n-3 fatty acid is markedly higher, reflecting the limited intake of long-chain n-3 PUFAs [33,34]. Consequently, the Omega-3 index value (eicosapentaenoic acid (EPA) + docosahexaenoic acid (DHA) percentage in erythrocyte fatty acids) is far below the recommended value of 8% in the US population, while the corresponding value for Japanese is well above 8% [35,36]. These differences in lipid intake should be considered when evaluating the effect of egg consumption on circulating cholesterol and its distribution in different lipoproteins. In addition to the differences in dietary lipid profiles, the different proportion of the source of dietary protein should also be taken into account since animal protein, in general, tends to elevate blood cholesterol levels [37,38], and vegetable protein intake was inversely associated with cardiovascular mortality as demonstrated in Japanese [39]. The more important point that should be emphasized is the combined effects of all nutrients [40]. Indeed, in a pooled analysis of prospective cohorts among US adults, higher consumption of dietary cholesterol or eggs was significantly associated with a higher risk of CVD and all-cause mortality in a dose–response manner [41]. This news confused Japanese consumers, although the paper described that generalizing our results to non-US populations requires caution due to different nutrition and food environments and chronic disease epidemiology. Regarding dietary cholesterol and CVD risk, the American Heart Association Science Advisory also emphasized the recommendation that gives a specific dietary cholesterol target within the context of food-based advice challenging for clinicians and consumers to implement; hence, a guide focused on dietary patterns is more likely to improve the diet quality and promote cardiovascular health [42].

As shown in Figure 1, the amounts of cholesterol consumed by Japanese are slightly higher compared to those by the US population. However, a distinct difference in a food source contributes to cholesterol intake between the two countries. In Japan, the egg contributes to approximately half of the cholesterol consumed, while meat contributes to less than half of the egg (18%). Seafood is the second main contributor in Japan. In contrast, meat is the largest contributor in the US at 37%, followed by eggs at 25% [13]. When considering the nutrient compositions of these foods, the differences in dietary cholesterol sources may delicately influence the response of blood cholesterol level to dietary cholesterol. However, the association between red and processed meats or eggs and stroke risk is not clear [38,43]. It seems more important to consider all foods consumed. Indeed, an optimal intake of whole grains, vegetables, fruits, nuts, legumes, dairy, fish, red and processed meat, eggs, and sugar-sweetened beverages showed an important lower risk of CHD, stroke, and heart failure [44].

Egg contributes to a relatively low amount of SFA, approximately 1.5 g per egg (~50 g), while beef on average contains more than three times of SFA than an egg per 50 g. When calculating the actual amounts of meat consumed by the US population, the effect of SFA should be considered. One of the most important recommendations in the current worldwide dietary guidelines is to consume as low SFA as possible by replacing them with unsaturated fatty acids, particularly PUFA [45,46,47], though there are still objections to this recommendation [48]. It has been suggested that not all SFAs are bad for the health, particularly those in some dairy products [49,50]. The contribution of cholesterol and SFA in dairy products between Japan and the US is not clear, although the US population consumes them more than twice as much.

The difference in the amounts and composition of nutrients and foods consumed between Japan and the US population may cause a difference in the mortality pattern. There appears to be no robust evidence to avoid egg consumption, as consuming at least one egg daily in Japan may not be similar to the US population. This point will be discussed in the section on T2DM.

Kim and Campbell suggested that the dietary cholesterol in a whole egg is not well absorbed, which may provide mechanistic insight why it does not acutely influence plasma total-cholesterol concentration and is not associated with longer-term plasma cholesterol control [51]. It is also possible that the quantity and quality of nutrients and non-nutrient components (such as plant sterols and polyphenols) contained in foods influence egg cholesterol consumption. Homeostatic regulation of cholesterol synthesis is also considered. The intake of three eggs daily appears to regulate the endogenous synthesis of cholesterol in such a way that the LDL-cholesterol and HDL-cholesterol ratio is maintained [52]. More information is, therefore, needed to understand the relationship between egg consumption and blood cholesterol level and risk of CVD. To solve this mystery, it is essential to understand the intestinal absorption of cholesterol from eggs and nutrients and cholesterol interactions in eggs [53].

It is clear that type of diet is one of the most important factors for modifying the effect of dietary eggs on CVD risk and hence, healthy life. Even though there is no difference in the response of blood cholesterol level to egg consumption between the Japanese and US populations, the consequence of CVDs may differ considerably, requiring more attention to the dietary pattern. Keeping the dietary cholesterol below 300 mg daily or as low as possible is an iron rule in the US [54]. Since the contribution of cholesterol by one egg is less than two-thirds of this limitation, daily consumption of one egg may not cause risk for CVD even in the US population if they improve their healthy eating index score (Dietary Guidelines for Americans, 2020–2025, the score is 56 in age 19 to 30). Because of the distinct difference in the dietary pattern, it seems plausible that the confidence toward egg consumption is much higher in Japanese than in the US population at present. Egg as a protein-rich food may improve vascular function [55] and effectively reduce mortality risk and promote longevity in old ages above 80 years [56]. Egg intake under extreme undernutrition during the childbearing period plays a critical role in preventing nonfatal coronary events during the postmenopausal period in Chinese women, suggesting not to limit egg intake [57]. In this context, eggs are cost-efficient in delivering nutrients even in the American diet [58].

## 4. Egg Consumption and Health: Global View

### 4.1. Cardiovascular Diseases

In addition to global observation and intervention studies reporting egg consumption and blood cholesterol levels, and CVD risk, there are many systematic reviews and meta-analyses, including umbrella or Mendelian randomization analyses (mostly since 2018, [3,4,26,41,44,55,57,59,60,61,62,63,64,65,66,67,68,69,70,71,72,73,74,75,76,77,78,79,80,81,82,83,84,85,86,87,88,89,90,91]). These studies did not necessarily show a consistent conclusion to egg consumption, but in almost all the observation studies risk of CVD or risk markers did not increase after moderate egg consumption, up to at least one egg daily [26,41,44,57,59,60,61,62,63,64,65,66,67,68,69,70,71,72,73,74,75,76,77,78,79,80,81,82,83,84,85,86,87,88].

In three large international prospective studies, including approximately 177,000 individuals from 50 countries in six continents, there were no significant associations between the moderate egg intake and blood lipids, mortality, or major CVD events [59]. In contrast, a study in US adults shows that higher consumption of eggs was significantly associated with a higher risk of CVD and all-cause mortality in a dose-responded manner [41]. In addition, there was a slightly elevated risk in a daily intake of one or more eggs among US veterans [60] and postmenopausal women [92].

The upper limit of egg consumption is not clear [61], although the meta-analysis showed a J-shaped association between egg consumption and stroke risk [91]. In a series of studies in China, egg consumption results in healthy responses and in one study, egg consumption is associated with lower total mortality [91]. However, consumption of cholesterol from non-egg sources may be detrimental to longevity, thus recommending habitual egg consumption, but not excess from non-egg sources [63]. In another study, among half a million Chinese adults, a moderate level of egg consumption (up to <1 egg daily) was significantly associated with a lower risk of CVD, largely independent of other risk factors [89]. Another Chinese cohort study also showed that eating one egg each day is not associated with an increase in CVD or all-cause mortality [63]. Although it is not clear whether these Chinese observations can be duplicated in Japanese, there may at least have similarities. However, the statement that can be considered safe for most populations should be handled with caution since the findings are observational in nature. It is also difficult to propose a healthy quantity or upper limit of daily egg consumption, although many studies showed that one egg’s daily consumption is not harmful to CVD events or risk markers. In this context, egg consumption was associated with a higher BMI and waist circumference, although it did not influence LDL-cholesterol or C-reactive protein in US older adults [64].

On the contrary, in intervention trials, the blood cholesterol level tends to moderately elevate by egg consumption in a dose-dependent manner [65]. In this respect, RCTs with long-term follow-up are needed to guarantee the association between egg consumption and human health.

In a meta-analysis of 23 prospective studies, higher consumption of eggs (more than one egg daily) was not associated with an increased risk of CVD but associated with a significant reduction in risk of coronary artery disease [66]. In addition, in three large international prospective studies from 50 countries in six continents, there were no significant associations between egg intake and blood lipids, mortality or major CVD events [59]. An umbrella review of observational studies showed no significant association between egg consumption and numbers of health outcomes, including cancer, cardiovascular and metabolic disorders, and evidence of possible beneficial effects toward stroke risk [67].

### 4.2. Stroke and Hypertension

The association between egg consumption and risk of stroke also appears complicated, from no association to lowering risk [43,62,63,68,69] or increasing risk [70,90]. A higher risk was associated with hemorrhagic stroke more than with ischemic stroke [68]. Although the dietary pattern of fish and fruit intake correlated with lower hypertension in China, the egg was also a contributor (egg intake ≥150 g/week; adjusted odds ratio, OR = 0.88; *p* < 0.29) [71]. Higher egg consumption was attributed to a reduced probability of stroke in Asia (RR = 0.83) but not in North America (RR = 0.95) or Europe (RR = 1.02) [91]. In the meta-analysis of prospective cohort studies, egg consumption was associated with a lower risk of hypertension (adjusted RR = 0.79, *p* = 0.001) [72], while in a study with Australian adults, lower intake of the egg together with other foods resulted in higher systolic blood pressure [73]. Egg and cholesterol intakes were associated with a higher risk of hypertension in French women [74]. The inconsistency may depend on the amounts of eggs consumed.

A recent dose-responsive meta-analysis of prospective studies showed no significant association between egg consumption and stroke risk (RR = 0.92), but the subgroup analysis resulted in a different association among geographic locations [91]. A decreased risk was observed for the intake of one to four eggs weekly and an increased risk for the intake of more than six eggs weekly. The different responses among countries may partly be explained by the amount of dietary SFA. Although SFA consumption increases the risk of stroke in the Western populations, this trend was reversed in Japanese due to a relatively low amount of SFA consumption in Japanese, and the risk tended to be lower with the increasing SFA intake [93,94]. From this situation, consumption of 20 g (<10% E) of SFA daily is recommended to be optimum in Japanese [95]. The specific response patterns in Japan may at least have a connection with a lower risk of T2DM relative to Western countries [4,96].

At present, there is no robust evidence adaptable to all populations regarding the role of egg in blood cholesterol level and CVD risk in both prospective and intervention studies. Probably because of the limited support to nutrition research that makes it difficult to conduct human nutrition research with many interesting topics, it seems unlikely to obtain a perfect answer to this question. The latest report on the meta-analysis of RCTs shows that more egg consumption in a long time may lead to cause hazardous blood cholesterol distributions, though RCTs with long-term follow-up are needed to guarantee the association between egg consumption and human health [75]. Thus, the answer to the question remains still unsolved yet. However, in the meta-analysis of prospective studies, the response depends on the amounts of egg consumed. The stroke risk decreases in less than one to four eggs weekly and increases in more than six eggs weekly [91]. In this context, it is interesting to know how hyper-responders respond to egg consumption. Compared to 0 eggs, hyper-responders had significant increases in plasma total, LDL- and HDL-cholesterol (*p* < 0.001) with no changes in the LDL/HDL ratio following intake of 3 eggs/day. They also had higher concentrations of large LDL (*p* < 0.01) with no changes in small LDL. Interestingly, both hyper- and normal responders had significant increases in large HDL particle concentration (12%) and plasma lutein (17%), zeaxanthin (30%), and choline (12%) (*p* < 0.001) compared to 0 eggs [90]. These results may support the safety of egg consumption even in hyper-responders. The increased level of blood carotenoids by egg consumption was also observed in Japanese [29].

## 5. Egg and Type 2 Diabetes Mellitus

The relationship between egg intake and risk of T2DM is also complicated. In observation studies reported from several countries, no association is observed between egg consumption and T2DM in healthy persons or no hazardous in T2DM patients [3,4,9,97,98,99,100,101]. In some cases, egg consumption lowers the risk of T2DM or ameliorates glycemic control [96,102,103,104,105,106,107]. In contrast, several studies indicate a positive association between egg consumption and T2DM risk [92,98,108,109,110].

The observation study in Japanese showed a decreasing trend of >20% of T2DM risk in women with the highest quartile of egg intake, but not in men [102]. Dietary reference intake for Japanese 2020 indicates that there is no association between the number of eggs consumed and the prevalence of T2DM. Moreover, there is no association between egg consumption and risk for CHD, even in DM patients. In contrast, a series of studies intended for the US population indicate a positive association between egg consumption and risk for T2DM [92,98,99,109]. For example, in the meta-analysis of cohort studies, egg consumption was associated with incident T2DM (RR/egg/day = 1.13), and it was stronger for studies conducted in the US (RR = 1.47), while results were null for studies conducted elsewhere [98].

In other studies, daily consumption of one egg was associated with higher T2DM risk among US studies (relative risk, RR = 1.18), but not among European (RR = 0.99) or Asian (RR = 0.82) [4]. As shown in Figure 4, replacing one whole egg per day with one serving of yogurt, whole grains, nuts, reduced-fat milk, high-fat cheese or full-fat milk was associated with a 9–19% lower risk of T2DM, while replacement with legumes, potatoes, refined grains, poultry, low-fat cheese, unprocessed meat, fish, or processed meat is not associated with T2DM [4]. In the US population, eggs are often consumed with red or processed meat, refined grains, and sugary beverages. Therefore, heterogeneity in the response may be attributable to dietary patterns [4,105]. The same trend was observed in Korean T2DM patients, and higher egg consumption increased the risk for CVD, thus providing a basis for the development of an optimal dietary cholesterol intake guideline for the Korean population [109]. In middle-aged and elderly Chinese people, the association between egg consumption and T2DM was nonlinear, and higher egg consumption was not associated with an elevated risk for T2DM [101]. However, another Chinese study showed higher egg consumption is associated with an increased risk of diabetes and with the same amount of intake, women were at an increased risk than men [110]. These contrasting results from the same country may indicate that the difference in the dietary pattern may be a crucial factor influencing the association between egg and DM risk markers, including blood cholesterol level. Considering the regional difference in dietary patterns, it is plausible that there is a large difference in T2DM distribution in mainland China [111]. People can safely consume up to seven eggs per week without T2DM and in patients with established T2DM but with a healthy lifestyle. In this context, eating white potatoes with eggs appears to control glycemic response [112].

There is an increasing prevalence of obesity and DM in the US during these decades [113,114]. The different responses to egg consumption between the US and other countries are probably attributable to residual confounding by dietary behaviors or food preparation methods restricted to certain populations. In this context, the influence of frequent consumption of meats by the US population is suggested. Since replacing red meat consumption with other protein sources, including egg, was associated with a lower risk of T2DM risk [115], it is suggested that the purported egg-T2DM risk relation in the US population may be biased due to failure to investigate egg-meat interactions [100]. However, since prospective analysis did not show an association of egg intake with incidence of T2DM among African Americans despite positive relation of egg consumption with prevalent T2DM, further investigations are needed to ascertain T2DM risk among non-meat eaters with high egg intakes or between African Americans and Native Americans [97].

Another point that should be considered is a high prevalence of obesity in the US, and the proportion of overweight is around 40% of the total population compared to 4.5% in Japanese. [116]. t has been pointed that a higher intake of saturated fats associates with the risk of obesity [113] and consequently elevates the risk of T2DM [117]. However, the possibility cannot be disregarded that obesity may not directly be connected with T2DM risk.

There is a claim that insofar as a diet is consistent with dietary guidelines, consumption of 6 to 12 eggs per week has no adverse effect on major CVD risk factors in individuals at risk of developing DM or with T2DM, except for the US population [99]. A healthy diet based on population guidelines, including more eggs than currently recommended by some countries, may be safely consumed by people with prediabetes or T2DM, probably except for the US population [100]. A more recent report of a meta-analysis of an individual-based cohort study based on the US National Health and nutrition examination survey (NHANES), egg intake had no association with CHD and total mortality and is associated with a lower risk of mortality from stroke, while associated with T2DM, hypertension, C-reactive protein, and markers of glucose/insulin homeostasis [62]. It is interesting to know whether the same is true for the Canadian population, whose dietary habits may not be greatly different from that of the US population, but the obesity rate is considerably lower, at approximately 28%. A possibility exists that Americans may have a specific physical constitution prone to develop T2DM.

The mechanism responsible for the inhibition of T2DM onset by the egg has been proposed. Egg consumption may improve factors associated with glycemic control and insulin sensitivity in adults with pre-T2DM and incident T2DM [103]. However, the mechanism for preventing T2DM pathogenesis by eggs is not totally recognized in all related areas of diseases, and egg consumption remained a plausible risk factor for T2DM [97]. In the meta-analysis of international interventions, egg consumption reduced risk for T2DM, and it is speculated that the egg-derived peptide may take part in such function [105]. Egg consumption was inversely related to the risk of incident T2DM in men but not in women, indicating gender differences in the relationship between egg consumption and disease risk. Numbers of food-frequency ranks showed both temporal trends and associations with gestational diabetes, but the frequency of egg consumption (negative temporal trend) is notable as it was also positively associated with the insulin disposition index. Such factors may have contributed to the observed temporal trend in gestational DM risk, but the overall detectable effect appears to have been small [106]. At any rate, many cases of T2DM could be prevented with lifestyle changes, including maintaining healthy body weight, consuming a healthy diet, and staying physically active.

Choline is listed up in US dietary guidelines as an essential nutrient, but not in Japan, and egg, milk and meat are the major sources of choline not only in the US but in Western diets. Replacing red meat consumption with other protein sources, including egg, was associated with a lower risk of T2DM in US cohorts [114] (refer section of “Debates around Trimethylamine Oxide”). However, it is not clear at present whether this mechanism is also responsible for cases in other Western countries.

The possibility that the products of the gastric microbiome may participate in cardiomyopathy due to DM is a clue to the answer for the specific response of the US population to T2DM risk [115,117]. The association between higher blood trimethylamine (TMAO) concentrations and increased risk of major adverse cardiovascular events and all-cause mortality has been reported, thereby opening some avenues on the role of dysbiosis in cardiovascular risk in T2DM patients [118]. In addition, it has been shown that there is a reciprocal interaction between the gut microbiota and n-6 and n-3 PUFAs that preferably modulate the microbiota population to a proportion favorable for health [119]. The US-specific response can also partly be explained by a significant difference in the n-6/n-3 ratio of dietary fats between the Japanese and the US populations.

In summarizing, the relationship between consumption of eggs and the risk of T2DM from observation studies is not consistent, while intervention studies provide promising evidence that consuming eggs ameliorates T2DM risk. In this context, some egg components and egg-derived peptides may be beneficial in this context of T2DM in terms of insulin secretion and sensitivity, oxidative stress, and inflammation [106]. The different dietary patterns may explain the US-specific response, but the heterogeneity of the associations among US, European and Asian cohorts reflect differences in consumption of eggs habits warrants further investigation [4]. Furthermore, more systematic research is necessary to differentiate eggs, T2DM risk, and CVD risk in diabetics before drawing firm conclusions.

## 6. Debates around Trimethylamine Oxide (TMAO)

A part of ingested choline is transformed to trimethylamine (TMA) by intestinal microbes and then absorbed into the liver, where it is oxidized to trimethylamine-N-oxide (TMAO) by the flavin-containing monooxygenase enzyme 3 (FMO3). TMAO is then transported to tissues or excreted into the urine. TMAO acts atherogenic [120,121,122,123,124]. Updated systematic review and meta-analysis indicate that TMAO increased the hazard ratio (HR) of CHD mortality [125]. Plasma TMAO levels can be increased with healthy and unhealthy diets and do not correlate with the extent of atherosclerosis but with plaque instability [126]. There is also a comment that patients with impaired renal functions, including the elderly, should also avoid egg yolk to avoid the influence of TMAO [127].

Phosphatidylcholine (PC) is the major form of choline in foods, such as dairy products, eggs, and meats. Therefore, it has been speculated that consumption of foods high in PC may correlate to CHD risk, although higher concentrations of circulating TMAO cannot simply be interpreted as a marker of unhealthy food intake or dietary pattern [128]. Since the extent of transformation of choline to TMA depends on the type of gut microbiota (*Firmicutes and Proteobacteria*), the response pattern to dietary egg discords in several studies, including an increase or non-increase in blood TMAO levels [129,130,131,132]. In a population-based cohort of Japanese people, no significant association was observed between choline and betaine consumption (one of the precursors of TMA) and the risk of cardiovascular mortality [133]. The amount of PC consumed daily by Japanese people is reported to be 2.17 g (approximately 300 mg as choline), and the eggs were major contributors to PC intake in that study [134]. Choline intake in US adults is reported to be around 350 mg daily, and meats are the main source [135].

To understand this complicated issue, it is important to know the extent of the transformation of choline to TMA in the intestine. As specific types of gut microbes participate in the transformation of choline and related substances to TMA, the rate will be influenced by the type of diet, resulting in a considerable individual difference. First, the concentration of serum TMAO is either dependent or independent on the amount of egg consumed [131,132]. In the intervention study with healthy Americans, intake of up to three eggs per day did not increase plasma TMAO [136]. In another study in a healthy US population, consumption of eggs was significantly associated with increased plasma TMAO concentration when egg intake was assessed with a semiquantitative food-frequency questionnaire, but not with repeated 7-dietary records, in contrast to the positive association in both cases with fish [128]. It has been shown that the plasma TMAO concentration is also dependent on the type of fish [137]. In the cohort study in China, the significant association has been observed between consumption of fishes or egg and the urinary TMAO level (β = 0.17, *p* < 0.0001 or β = 0.07, *p* = 0.08, respectively) [138]. At present, egg consumption is considered not to associate with plasma levels of TMAO, but the possibility of positive association cannot be excluded, probably depending on the amounts of egg consumed. Rather, plasma TMAO concentration is more strongly affected by intraindividual variation, probably reflecting the difference in gut microbiota composition [139].

However, it is plausible that the phospholipid type of choline may not easily be transformed to TMA in the small intestine than the free forms of choline since PC is relatively smoothly absorbed. This assumption was confirmed by comparing the dietary PC and choline bitartrate, as shown in Figure 5 [140]. This study also emphasized that PC should be avoided due to the large individual difference in TMAO-producing characteristics. Meat-rich diets in the US population may, in part, be related to the association between consumption of eggs and T2DM since meat is the major source of TMAO [135].

From these complex situations, the extent of CVD risk that is attributable to TMAO may also differ individually [124,140]. A higher circulating concentration of TMAO may not simply be interpreted as a marker of unhealthy food intake or an unhealthy dietary pattern. The causal relations between diet, TMAO, and disease risk seem complex and still not dissolved well. A study targeting microbiome metabolites and CVD prevention and therapy are also reported [141]. Under such circumstances, there may be a new opportunity for functional food development [142], and these approaches will provide a concrete countermeasure for anyone who can eat eggs. Regulating TMAO production and associated gut microbiota may become a promising strategy for antiatherosclerosis therapy [143]. These approaches from different viewpoints may contribute to eating eggs safely unless there is no egg allergy risk. However, since some fish are the direct source of TMAO [144], consuming such fish must be considered in the Japanese population. These observations may facilitate the adoption of an individualized nutrition-based approach to TMAO at least [145]. Recent findings regarding the circulating TMAO-lowering effects of specific foods, food constituents, and phytochemicals may also be useful to diminish untoward side effects expected to egg [124].

## 7. Cholesterol-lowering Components in Egg

Egg white protein is the only component of eggs that has been tested by human trials in terms of reducing serum cholesterol levels. In a trial among Japanese female college students, a decrease of serum total- and LDL-cholesterol concentrations were noted by the daily intake of 23 g of egg whites for 4 weeks as a protein source compared with cheese (milk protein) [146]

Several mechanisms have been reported for the cholesterol reduction potential, including the enhancement of cholesterol catabolism and suppression of very low density lipoprotein (VLDL) release from the liver by a sulfur-containing amino acid contained abundantly in egg whites [147,148], as well as suppression of intestinal cholesterol absorption by ovomucin in egg whites [149]. Subsequent studies reported a key mechanism of the cholesterol-lowering effect of egg white, in which the physicochemical properties of ovalbumin and ovotransferrin in egg whites interferes with the dissolving of dietary cholesterol in bile acid micelles and consequently suppress the absorption of lipids, including cholesterol [5,150].

However, consumption of 23 g daily of egg whites would be difficult because of their specific flavor and physical properties. The flavor was improved by lactic acid fermentation technology without influencing egg whites’ function [151]. The function of egg whites was also maintained in the product [152]. In the feeding trial with lactic acid-fermented egg whites for 8 weeks in adult males of normal cholesterol concentration (229 ± 1.6 mg/dL), there was a dose-dependent decrease in serum total-cholesterol concentration from the baseline, and the difference was significant in the dosage of 8 g per day, −11.0 ± 3.7 mg/dL. LDL-cholesterol also decreased significantly in this dose, −13.7 ± 3.1 mg/dL (*p* < 0.05) [6]. Thus, lactic acid-fermented egg whites can be used as a functional food.

In addition to the role of egg whites in cholesterol-lowering, animal studies indicate that egg phospholipids may also lower blood cholesterol by interfering with cholesterol absorption [7,153]. The available information thus supports the view that several components of the egg may control cholesterol metabolism.

## 8. Rebuttal to “Innocent Theory” of Egg Consumption

Due to the disharmony among researcher viewpoints, at present, there is controversy concerning the consumption of eggs and health, and several objections have been raised against the healthy theory of eggs. For example, there is a belief that the consumption of dietary cholesterol and egg yolks is harmless is a misconception, not only by the public but also by physicians. Dietary cholesterol, including egg yolks, is harmful to the arteries, and patients at risk of CVD should limit their cholesterol intake. Stopping the consumption of egg yolks after a stroke or myocardial infarction would be like quitting smoking after a diagnosis of lung cancer: a necessary but late action [154], and therefore, appropriate education is indispensable for a right judgment [155,156].

Cholesterol management is pointedly important to DM patients in the US, and the American Heart Association stresses that patients with dyslipidemia, particularly those with diabetes DM or at risk for heart failure, should be cautious in consuming foods rich in cholesterol [42]. The Guidelines for Prevention of Atherosclerosis by Japan Atherosclerosis Society 2017 also describes that consumption of foods containing large amounts of eggs should be avoided to improve lifestyle and recommends consumption of <200 mg daily of cholesterol, particularly when there is the risk for atherosclerosis. From the standpoint of managing the atherosclerotic disease, it is necessary to avoid looking away from this point. Since there are several limitations in evaluating epidemiological studies, it seems reasonable that further studies are warranted to evaluate whether such conclusions can be extrapolated to all populations as dietary patterns are difficult to disentangle. Until more definitive data are available, future guidelines should stress the importance of a well-balanced diet and its inextricable part in a healthy lifestyle, albeit without strict limitations. Whether eggs can or cannot be consumed is not an unanswered question, but the question is how much egg can be consumed in healthy persons, and beneficial effects of eggs could be expected when the diet is prudent, as repeatedly pointed out.

Despite these situations, the efforts to reduce the cholesterol content of eggs continue from at least two different standpoints, either the mechanical removal of cholesterol from eggs or cholesterol reduction in eggs by breeding. Among a variety of reduction technologies, extraction of cholesterol by critical carbon dioxide seems one of the excellent procedures [157]. Egg cholesterol can be reduced by feeding specific compounds, such as phytosterols or polyphenols, to laying hens, but the extent of reduction is approximately 10% [158,159], except for the 25% reduction of cholesterol in one report using special Chinese medicinal herbs [160]. However, both approaches seem not cost-effective at present.

## 9. Conclusions

A systematic judgment from a nutritional background is essential to understand the association between consumption of eggs and blood cholesterol level and hence, CVD risk. Although information from the intervention studies on egg and blood cholesterol levels is limited in Japan, it seems plausible that the daily consumption of one egg is uninfluential to the blood cholesterol level. Among multiple confounding factors involved in this issue, critical attention should be paid not only to nutrients other than cholesterol in the egg but also the dietary pattern. Several components of eggs may also interfere with the rise of blood cholesterol [5,6,7,51].

While consuming relatively more eggs than in the US population, Japanese people are carrying on healthful eating habits and spending an ultra-aging society with respect to the risk of noncommunicable diseases. It seems likely that daily consumption of more than one eggs for a long period will result in the elevation of blood cholesterol level, irrespective of LDL- or HDL-cholesterol, as suggested in the recent systematic review and meta-analysis of 17 RCTs [75]. In this context, RCTs with long-term follow-up are essential to guarantee the association between consumption of eggs and health. At present, eggs are an indispensable healthy and cost-efficient food for residents of under developing countries and even for the public in developed countries [58,161]. When considering the amounts of egg consumed in Japan, it seems likely that we are receiving the considerable benefit of egg intake as a source of various important nutrients. The American Heart Association Science Advisory is stressing the recommendation that gives a specific dietary cholesterol target within the context of food-based advice is challenging for clinicians and consumers to implement; hence, guidance focused on dietary patterns is more likely to improve diet quality and to promote cardiovascular health [42]. The egg is an indispensable source of various nutrients, in particular dietary protein, for healthy living, but it will be profitable only under healthy diets. The Japanese dietary pattern, Washoku, may be sufficient to avoid the influence of cholesterol by consumption of eggs. The correlation between consumption of eggs and cancer may also be a matter of concern, though not conclusive [162,163].

Understanding the difference in the individual response of blood cholesterol to the consumption of eggs, including the gene variation in hyper-responder to dietary cholesterol, is the key to comprehend the association of egg and cholesterol at the personalized level. At present, however, approaches from nutrition sciences are indispensable.

## Figures and Tables

**Figure 1 foods-10-00494-f001:**
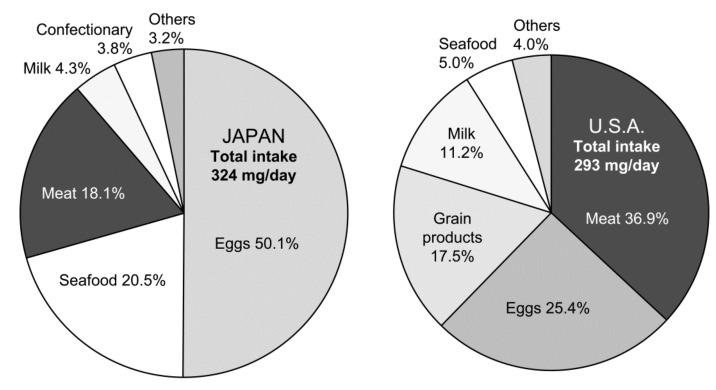
Dietary sources of cholesterol in Japanese and US populations. Japanese data are from the National Health and Nutrition Survey, 2017 (Ministry of Health, Labor and Welfare, Japan), and the US data are from the National Health and Nutrition Examination Survey, 2003–2014 [13].

**Figure 2 foods-10-00494-f002:**
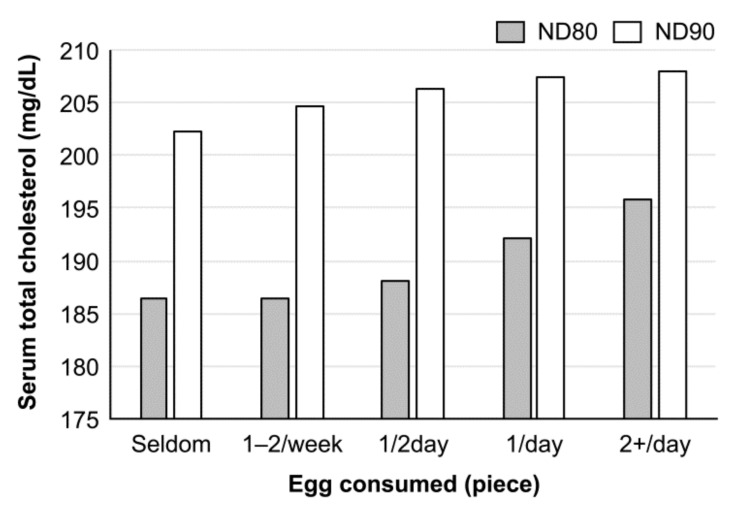
Relationship between egg consumption and serum cholesterol level in Japanese women (observation studies). Cholesterol levels were adjusted with age, body mass index (BMI), smoking, and alcohol intake. There was a significant association between egg consumption and serum cholesterol levels in NIPPON DATA 80 (*p* < 0.001), but the association disappeared in NIPPON DATA90 (*p* = 0.63). In both studies, the association was not observed in men. The order of egg consumption was reversed from the original figure in reference [19,20,21].

**Figure 3 foods-10-00494-f003:**
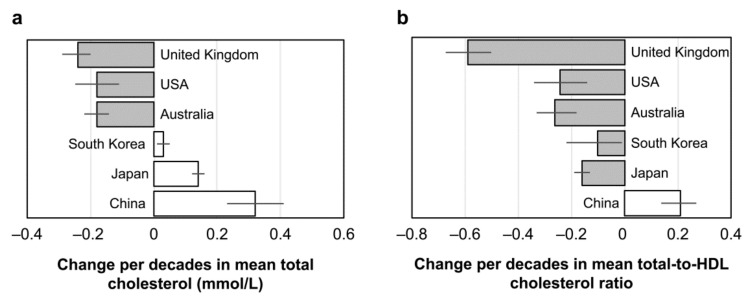
Change per decade since ~1980 in mean total-cholesterol level (**a**) and mean total-to-HDL-cholesterol ratio (**b**) by male aged 40–59 years. Values are means with 95% confidence intervals. Similar but more clear changes are observed in females. Extracted data from reference [30].

**Figure 4 foods-10-00494-f004:**
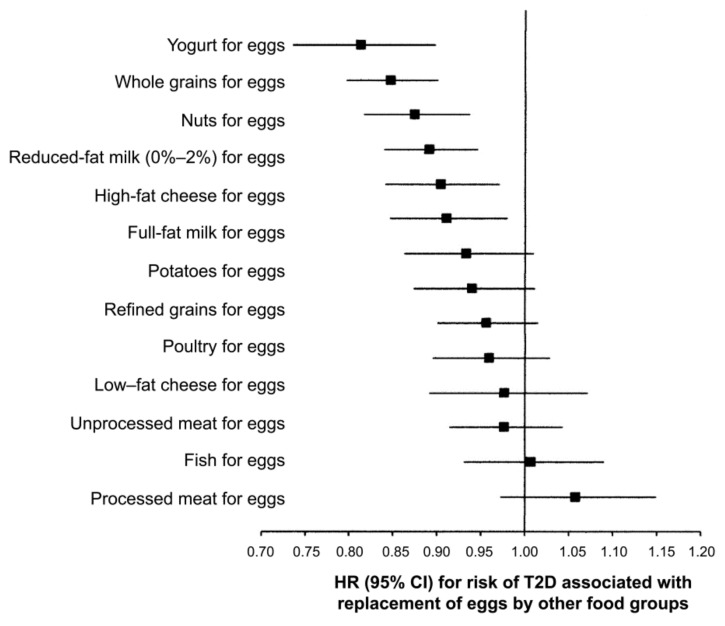
Statistical model-based hazard ratios (HRs) and 95% CIs for incident type 2 diabetes mellitus (T2DM) associated with replacing one egg/day with one serving/day of other foods in the NHS, the NHS II, and HPPS (pooled analysis, *n* = 213,798) [4].

**Figure 5 foods-10-00494-f005:**
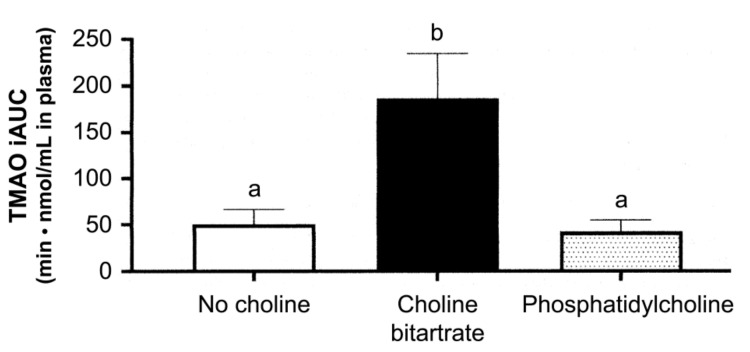
Plasma trimethylamine-N-oxide response to different meal consumptions as the incremental area under the curve (iAUC) across 6 h after ingestion. Different letters show a significant effect of the study meals (one-way) ANOVA. Values are mean ± SE, *n* = 37 per study. Different letters, *p* < 0.01 [140].

**Table 1 foods-10-00494-t001:** Comparison of daily nutrient intake between Japanese and the US population.

	Energy	Protein g (E%)		Lipids g (E%)		Carbohydrate	SFA	MUFA	PUFA g (E%)		Chol
	(kcal)	Total	Animal	Total	Animal	g (E%)	g (E%)	g (E%)	n-6	n-3	mg
Japan	1930	71.8	39.3	61.0	31.8	254.0	17.59	22.80	10.75	2.51	340
		(14.9)		(28.4)		(52.6)	(8.20)	(10.6)	(5.01)	(1.17)	
USA	2155	81.9	52.4	88.2	49.9	248	28.7	30.2	18.6	2.2	307
		(15.2)		(36.8)		46.0	(12.0)	(12.6)		(8.69)	(0.92)

Data are males and females age ≥20 from the Japan National Health and Nutrition Survey, 2018 (Ministry of Health, Labour and Welfare, Japan) and US National Health and Nutrition Examination (National Center for Health Statisitics, USA). Survey, 2017–2018. Proportions of the animal source of protein and lipids are 54.7% and 52.1% for Japanese and 64.0 and 56.6 for the US population, respectively. Dietary fiber intake is 15.0 g (Soluble 3.5 g and insoluble 10.8 g) and 16.9 g for Japanese and the US population, respectively. The ratio of n-6/n-3 PUFAs is 4.3 and approximately 10 for Japanese and the US population, respectively, reflecting the large difference in fish intake, which is 70.1 and 18.0 g/day, respectively.

## Data Availability

Data presented are from publicly available online databases.

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
