# Peer review of "Nutritional Viewpoints on Eggs and Cholesterol"

_foods, 2021, doi:10.3390/foods10030494_

Round 1
Reviewer 1 Report
In this interesting review Sugano M. et al. discuss the effect of egg consumption on blood cholesterol levels and cardiovascular risk comparing different dietary patterns. This paper could be of interest for this journal but there are some issues which need an extensive revision:
- English language needs an extensive editing;
- In the abstract section: from line 20 to line 25, this sentence is not clear. Please, make it easier and explain it properly.
- e. Line 64: “observantion”. There are several grammatical errors in the review, please review the whole manuscript and edit it properly.
- The review is exhaustive, but several sentences are excessively prolix and superfluous. Try to simplify and avoid ripetitions.
- Any correlation between egg consumption and cancer (i.e. colon cancer)?
- How does you define hyper-responders and normal responders to egg consumption? Are there some variants (SNPs) in genes involved in lipid metabolism and/or variants of the gut microbiota, known in literature, which may explain these traits? Please, comment.
Reviewer 2 Report
The report is an interesting comparison mainly between the Japanese and American research relating egg and cholesterol intake to health outcomes. However, I have some comments:
Major comments
The manuscript would greatly benefit from a more condensed way of writing and a significantly more structured layout. For example, under the section titled “Egg and type 2 diabetes mellitus”, there is text related to CVD. I would suggest that the authors clearly separate the text related to different outcomes to their own sections: e.g. CVD outcomes in one section and T2D in another. Also, in the CVD section, it would be useful to include subsections, which discuss about e.g. CHD and stroke outcomes separately. There is now quite a lot of repetition in the text and this would improve also that aspect.
Another issue that would greatly improve the readability is that the authors could include tables that describe the characteristics and major findings of the many reviews and meta-analyses that they refer to in the text.
Minor comments
Line 32. I wouldn’t say eggs are now considered a healthy food by a majority of experts. Neutral perhaps, but not a health food.
Line 46: reference needed for the sentence
Line 51: please clarify the “field of egg”.
Line 75-76: please clarify the sentence, its point is difficulty to understand.
Line 95: Please include a reference for the Japanese data
Line 115-116: please clarify the meaning of the sentence
Line 122: what is the reason that limiting egg intake is not possible?
Line 136: I guess these were not trials (?), so please rather use the word “studies”.
Line 148: Please clarify the sentence, its meaning is unclear.
Line 164: “more/less” remarkable?
Line 175: please clarify the confidence level. Also, please clarify what the “more healthy trends” means.
Line 180: This is quite a strong conclusion. Is the decrease in lipoprotein levels the only or even the main reason for decrease in mortality? what else has happen during the same time? Are there improvements in health care?
Lines 185-188: please add references for these statements.
Line 285: I would replace the words “preferably at least” with words “up to”, because there really is not enough data to indicate that long-term intake of more than 1 egg/d would be harmless.
Line 294: do the authors mean “safe egg consumption”?
Lines 295-296: please add references
Line 295: please check the ref 95 if that is incorrect, as it refers to T2D outcome.
Table 1: the numbers in parentheses are now somewhat confusing, because in the first row e.g. “Protein (g)” would indicate that the value in the parenthesis is grams, not E%. Therefore, I would suggest that you use e.g. style “Protein, g (E%)” in the first row. Also, there are some E% values missing in the table, e.g. for animal protein in the USA. Also, please add references to the studies mention in the first row under the table.
Line 383: “effect” should be reserved for experimental studies and not used with observational studies, so please use e.g. “no association with”.
Lines 355-357: This study did not find associations. This was also mentioned before in the text (lines 287-289).
Line 393: were there in fact differences in average egg intakes between these studies? Or could there also be some other differences between the populations that could explain the different associations?
Line 397: I’m sure the issue is not the limited interest, but more likely the limited funding for nutrition research that makes it difficult to conduct human nutrition research with many interesting topics.
Line 411: The ref 91 seems to be a conference abstract, so I would not cite it until the full paper is published.
Line 435: Is the figure from a published paper? It then needs a reference.
Lines 452-453: Is this the opinion of the authors or a finding in some previous studies?
Line 460: this is an example where the text is about a CVD outcome under T2D section. Please separate the outcomes to different sections.
Line 492: I think it is not generally accepted that more than 1 egg/day is safe, even if these authors in the cited paper might say so.
Line 519-520: another example where CVD outcome is mentioned under T2D section.
Lines 541-542: I don’t think there are intervention studies that would show amelioration of T2D risk, but rather T2D risk factors.
Lines 571-572: Is the amount of PC of 2.17 g correct? If eggs contribute 26% of this amount, that would be about 560 mg of PC from eggs. If 100 g of whole egg contains 210 mg of PC (USDA database of choline containing foods), 560 mg would be about 2.7 eggs. That seems like a very large average egg intake and most likely not correct.
Line 593: This sentence appears out of context.
Line 601: is meat really the major source of TMAO? Or do the authors mean that meat is the major dietary contributor to serum TMAO concentrations? Please clarify, because fish rather than meat has a high content of TMAO.
Line 615-616: It is quite far-fetched to say that anyone could eat eggs with those approaches. What about those allergic to egg?
Line 619-620: one must be very cautious with statements like this. It is very difficult to judge the health effects of a food just by looking at the content of a single nutrient or compound in fish, when the compound is not e.g. a poison that would be highly harmful. I don’t think there is any evidence that fish species with high TMAO content would be harmful or that any fish species should be avoided because of the TMAO in fish.
Lines 661-663: it is a bit unclear whether these statements are the viewpoint of the authors of the manuscript or those of the authors of the cited papers.
Line 678: I don’t think the question is that whether eggs can or cannot be consumed but rather HOW MUCH egg can be consumed safely.
Lines 682-683: this is quite a problematic statement, because that reference is an opinion written by known cholesterol denialists, not an actual research paper. It is strongly suggested that the authors at least include information about what is the consensus regarding saturated fat and cholesterol intake in FH patients.
Line 686-687: what does this sentence refer to?
Line 708: please clarify what “super-protracted-life society” means.
Line 715-716. What is this statement based on? I.e. what is the evidence that the Japanese are receiving benefit from egg intake?
Round 2
Reviewer 2 Report
The manuscript has been improved, but I still have a few comments for the authors to address. The line numbers below refer to the original version of the manuscript.
Line 32. I wouldn’t say eggs are now considered a healthy food by a majority of experts. Neutral perhaps, but not a health food.
Revised: As the result, the egg is now regarded as a healthy food from nutritional viewpoint.
-I still think it is too strong a conclusion to say that egg would now be regarded as a healthy food, based on all the evidence there is about egg intake and health (most of which show a neutral or harmful association, rather than a beneficial association). A “healthy” status would indicate that more is better, but in the case of egg intake that likely is not the case. A more appropriate would be to say e.g. that a MODERATE egg intake can be a part of a healthy diet.
Line 75-76: please clarify the sentence, its point is difficulty to understand.
However, many Japanese recognize that egg consumption results in elevation of blood cholesterol despite low mortality due to heart failure among developed countries at present.
-This sentence still is not very clear. What does the low mortality due heart failure in developed countries have to do with the first part of the sentence?
Line 122: what is the reason that limiting egg intake is not possible?
It is “inevitable”.
-I’m not sure if the word “inevitable” (meaning: certain to happen; unavoidable) is what the authors mean here. Please check.
Line 148: Please clarify the sentence, its meaning is unclear.
Thus, the results of hyper-responders were not reflected in the results and disregarded.
-This sentence is still unclear and perhaps also incorrect. I looked at the study and the state in the conclusions “Plasma levels of LDL1-C and HDL2-C were slightly but significantly increased by cholesterol ingestion in all subjects. However, those increases were not significant in any apoE group.” Please revise.
Lines 185-188: please add references for these statements.
These data can easily be obtained internet and reference may not be necessary.
-For an average reader it may not be easy, so please add the reference(s).
Lines 355-357: This study did not find associations. This was also mentioned before in the text (lines 287-289).
We are unable to find this in the manuscript.
-I apologize, I meant the lines 375-377 in the original manuscript, lines 376-379 in the current version. Please correct that statement as that study (Dehghan et al) did not find associations.
Line 393: were there in fact differences in average egg intakes between these studies? Or could there also be some other differences between the populations that could explain the different associations?
We described the possibility of different responses.
-Where have the authors described these different possibilities? I did not find it in the text.
Lines 571-572: Is the amount of PC of 2.17 g correct? If eggs contribute 26% of this amount, that would be about 560 mg of PC from eggs. If 100 g of whole egg contains 210 mg of PC (USDA database of choline containing foods), 560 mg would be about 2.7 eggs. That seems like a very large average egg intake and most likely not correct.
We think it should be “100 g of whole egg contains 210 mg of Choline” instead of “100 g of whole egg contains 210 mg of PC”. The original paper reported that 2.17g of PC was consumed per day (approximately 300 mg as choline intake). However, there is confusion in the estimated % contribution of egg. The amount of egg consumed was about 50 g (approximately 120 mg as choline) a day and it corresponded to about 40% of choline contribution (120 mg/300 mg×100). We revised as follows. “The amount of PC consumed daily by Japanese is reported to be 2.17 g (approximately 300 mg as choline) and the egg contributes, about 40% of choline as food [134].”
-According to the USDA choline database (https://data.nal.usda.gov/dataset/usda-database-choline-content-common-foods-release-2-2008), 100 g of whole, cooked, hard boiled egg contains 230 mg of total choline, of which 210 mg is phosphatidylcholine. Phosphatidylcholine intake (2.17 g) can’t be higher than the total choline intake (300 mg). I think the 300 mg in the cited paper by Shirouchi et al refers to free choline, not total choline, as they estimated it from the choline compounds they measured. Also, where does the 40% (26% in the original version) come from? I think it would be clearer to just state the eggs were the major contributor to PC intake in that study (Table 4 in Shirouchi et al).
